# Percutaneous Absorption of Fireground Contaminants: Naphthalene, Phenanthrene, and Benzo[a]pyrene in Porcine Skin in an Artificial Sweat Vehicle

**DOI:** 10.3390/toxics12080588

**Published:** 2024-08-13

**Authors:** Chandler Probert, Emma Nixon, R. Bryan Ormond, Ronald Baynes

**Affiliations:** 1Wilson College of Textiles, North Carolina State University, Raleigh, NC 27606, USA; 2College of Veterinary Medicine, North Carolina State University, Raleigh, NC 27607, USA; enixon@ncsu.edu (E.N.); rebaynes@ncsu.edu (R.B.)

**Keywords:** dermal absorption, polycyclic aromatic hydrocarbons, benzo[a]pyrene, porcine skin, fireground contaminants, firefighter

## Abstract

Firefighters face significant risks of exposure to toxic chemicals, such as polycyclic aromatic hydrocarbons (PAHs), during fire suppression activities. PAHs have been found in the air, on the gear and equipment, and in biological samples such as the skin, breath, urine, and blood of firefighters after fire response. However, the extent to which exposure occurs via inhalation, dermal absorption, or ingestion is unclear. In this study, three PAHs, naphthalene, phenanthrene, and benzo[a]pyrene, were applied to porcine skin in vitro in an artificial sweat solution to better gauge firefighters’ dermal exposures while mimicking their sweaty skin conditions using an artificial sweat dosing vehicle. Multiple absorption characteristics were calculated, including cumulative absorption, percent dose absorbed, diffusivity, flux, lag time, and permeability. The absorption of the PAHs was greatly influenced by their molecular weight and solubility in the artificial sweat solution. Naphthalene had the greatest dose absorption efficiency (35.0 ± 4.6% dose), followed by phenanthrene (6.8 ± 3.2% dose), and lastly, benzo[a]pyrene, which had the lowest absorption (0.03 ± 0.04% dose). The lag times followed a similar trend. All chemicals had a lag time of approximately 60 min or longer, suggesting that chemical concentrations on the skin may be reduced by immediate skin cleansing practices after fire exposure.

## 1. Introduction

Firefighters are repeatedly exposed to chemical hazards throughout their career. On the fireground, there are numerous types of chemical hazards to which firefighters can be exposed to, ranging from gaseous carbon monoxide and hydrogen cyanide to polycyclic aromatic hydrocarbons (PAHs) in the soot and smoke, and to formaldehyde during overhaul [1,2,3,4]. Several studies have now associated several types of cancers such as, but not limited to, multiple myeloma, non-Hodgkin lymphoma, prostate cancer, kidney cancer, and lung cancer to firefighters [5,6,7,8]. In 2022, the International Agency for Research on Cancer (IARC) determined that the occupational exposure of firefighting is a Group 1 threat, meaning it is a known carcinogen, based on “sufficient” evidence for cancer in humans [9]. The connection between firefighters and cancer is growing stronger as more evidence is generated. As a result, efforts to understand firefighters’ chemical exposures and identify methods to reduce, mitigate, or eliminate their exposures are increasing at a rapid rate [1,10,11,12].

There are three main routes through which firefighters can be exposed to toxic fireground contaminants: inhalation, dermal absorption, and ingestion. Inhalation is the most prioritized exposure pathway to protect against because of the lungs’ sensitivity and because several fireground contaminants have adverse respiratory effects [13]. Fortunately, exposure to chemicals can be drastically reduced when a self-contained breathing apparatus (SCBA) is worn properly during times of exposure [14]. The second route of exposure of primary concern is dermal absorption. This is the most difficult exposure pathway to assess. Researchers face many challenges, which include the need for a standardized collection media and method, biological differences and variability in exposure amongst individuals, and understanding the source of contamination, whether it be from the fire scene or from turnout gear that was contaminated previously and never thoroughly cleaned [15,16,17]. Due to these challenges, the dermal absorption of fireground contaminants is not well understood. Ingestion is the final route of exposure, where exposures are likely to occur in the living quarters of the fire stations eating contaminated foods [18].

Polycyclic aromatic hydrocarbons (PAHs) are a common fireground and environmental contaminant. Sixteen PAH compounds have been deemed as priority chemicals by the Environmental Protection Agency (EPA) based on their abundance, toxicity, and potential for exposure [19]. These compounds are produced as a byproduct of the incomplete combustion of organic material and can be produced naturally and anthropogenically. Exposure to PAHs has been associated with reproductive, developmental, and hemato-, cardio-, neuro-, and immunotoxicity effects [20,21]. Once absorbed into the body, PAHs can promote the formation of reactive oxygen species, which cause several adverse health effects. Additionally, a synergetic effect between dermal exposure to PAHs and UV radiation is thought to accelerate the development of skin diseases and cancer among researchers [22,23]. Although studies have primarily focused on PAH exposure through ingestion and inhalation, evidence related to the dermal absorption is slowly catching up. Animal studies using pig, guinea pig, rat, and monkey skin all show that PAHs can penetrate the skin and be absorbed into the body. Across several studies the absorption of lower-molecular-weight PAHs (2-3 rings) is generally greater than higher-molecular-weight PAHs (4+ rings) [24,25,26].

During fire response, it is almost impossible to entirely avoid exposures to polycyclic aromatic hydrocarbons due to their prevalence on the fireground. These compounds have been found in the air, on the outside and inside of firefighters’ turnout gear, and on the skin of firefighters after fire response activities [1,3,27,28,29,30]. The highest concentrations of PAHs have been observed in air samples, ranging up to 2700 µg/m^3^ outside the ensemble and up to 355 µg/m^3^ inside the turnout ensemble [30]. PAHs have been identified on the skin and have been found to be as low as <4.5 ng/cm^2^ and as high as 1200 ng/cm^2^ [1,31,32]. Increased concentrations of PAHs have been found in areas correlated with the interface areas of the turnout, for example, the neck, wrist, and forehead areas [33]. Additionally, several studies have shown that the level of exposure for individuals is variable and dependent on several factors; however, job duty (i.e., inside search vs. outside command) appears to play a significant role in the average level of chemical exposure [1,34].

This study aims to investigate the percutaneous absorption of polycyclic aromatic hydrocarbons through the skin during fire response. As mentioned previously, several studies have previously established that PAH compounds penetrate the skin; however, firefighters operate in extreme conditions, such as elevated temperatures, resulting in excessive skin moisture via sweating. The layer of sweat generated during fire response may interrupt dermal absorption mechanisms as PAHs become more lipophilic in nature as their molecular weight increases. To investigate how PAHs penetrate firefighters’ skin and the impact that sweat may play in dermal absorption, this study used an artificial sweat dosing vehicle. Three different PAHs, naphthalene, phenanthrene, and benzo[a]pyrene, were dosed to porcine skin in an artificial sweat dosing vehicle in an in vitro flow-through diffusion cell system. The three PAHs selected included a 2-ring, 3-ring, and 5-ring compound, which span the range of size and properties of EPA’s 16 priority PAHs. Ortho-phenylphenol was included as a control chemical since it is a known dermal penetrant. Porcine skin was used rather than human skin because it is more easily obtainable and has been shown to behave similar to human skin in absorption studies [35,36,37]. The goal of this study was to generate data to better understand the dermal absorption PAHs in human skin under fire response conditions so that the fire service industry can identify more effective protocols to reduce, mitigate, or eliminate firefighter exposures (the data are available in Appendix A).

## 2. Materials and Methods

### 2.1. Chemicals

Test chemicals ^14^C-naphthalene (specific activity = 57 mCi/mmoL), ^14^C-phenanthrene (specific activity = 55 mCi/mmoL), 14C-benzo[a]pyrene (specific activity = 26.6 mCi/mmoL), and ^14^C-orthophenylphenol (specific activity = 150 mCi/mmoL) were obtained from American Radiolabeled Chemicals (Saint Louis, MO, USA). Chemical properties of the radiolabeled chemicals can be seen in Table 1. Artificial eccrine perspiration (pH = 4.5 stabilized), an artificial sweat, was used as the dosing vehicle and obtained from Pickering Laboratories (USA). The cells used for the flow-through experiment were 9 mm (0.64 cm^2^) in-line diffusion cells obtained from PermeGear (USA). The collection media were created the day before the experiment and frozen until the day of the experiment. Ingredients for the collection media included bovine serum albumin fraction V (2.25% *w*/*v*), sodium chloride (0.3% *w*/*v*), potassium chloride (0.018% *w*/*v*), calcium chloride (0.014% *w*/*v*), potassium phosphate monobasic (0.008% *w*/*v*), magnesium sulfate (0.015% *w*/*v*), sodium bicarbonate (0.14% *w*/*v*), dextrose (0.06% *w*/*v*), distilled water (96.94% *v*/*v*), sodium heparin (0.25% *v*/*v*), amikacin (0.0063% *v*/*v*), and penicillin G sodium (0.0025% *v*/*v*) (Millipore Sigma, Saint Louis, MO, USA).

### 2.2. Flow-Through Diffusion Cell Set Up

In vitro studies have repeatedly reported lower absorption compared to in vivo studies across multiple species [26,38,39]. The underestimated values from in vitro studies often result from the removal of systemic uptake that occurs in the highly vascular and hydrophilic upper dermis following permeation through the stratum corneum. For this reason, the flow-through diffusion cell is preferred over the static diffusion cell system as it mimics the continuous uptake as in vivo experiments [40].

The flow-through diffusion cell system, described by Bronaugh and Stewart [41], was used to perfuse porcine skin membranes. Fresh porcine skin was obtained from the dorsal area of Yorkshire/Landrace pigs 20–60 kg in size. Pig skin has been shown to be similar to human skin with respect to stratum corneum, epidermal thickness, and permeability with less variability [42,43,44]. The pigs were shaved and dermatomed to a thickness of 200–300 µm with an electric dermatome (Padgett Instruments, Kansas City, MO, USA). Afterwards, each piece of skin was cut into a circular disk using a biopsy punch, placed into the diffusion cell, and secured in place, providing a dosing surface area of 0.64 cm^2^. The porcine skin membranes were dosed within 30 min of humane euthanasia of the porcine skin donor, so transepidermal water loss and transepidermal electrical resistance skin integrity tests were not performed [45]. However, leak tests and visual damage assessments were performed prior to the start of the experiment that may have occurred during the mounting procedure.

The dermal side of the skin disks was perfused with a bovine serum albumin collection media and maintained at a pH between 7.3 and 7.6. The temperature of the diffusion cells was maintained at 37 °C ± 1 °C using a heating block. The flow rate was maintained at 4 mL/h using a peristaltic pump. This flow rate was chosen to maintain sink conditions, ensuring that the concentration of the test substances in the receptor fluid remained low, thereby driving continuous diffusion through the skin. The flow rate of 4 mL/h is consistent with standard protocols and regulatory guidelines. The room temperature and relative humidity were recorded throughout the experiment for record keeping. Perfusate samples were collected in glass scintillation vials at 0, 15, 30, 45, 60, 75, 90, 120, 180, 240, 360, and 480 min. After the flow-through diffusion cell systems were set up, the chemical doses were added to each cell. The time for the experiment and sample collection started immediately after the last cell was dosed.

### 2.3. Dosing Procedure

Test chemicals ^14^C-naphthalene, ^14^C-phenanthrene, ^14^C-benzo[a]pyrene, and ^14^C-orthophenylphenol were made into separate dose mixtures. Each dosing solution was prepared by adding the test chemical to the artificial sweat and then adding acetone (1% of total volume). Each dosing solution was vortexed to ensure the test compound was thoroughly mixed. Skin disks were dosed with 100 µL of the respective dosing solution administered through a delivery channel using a pipette at the top of the cell, applying either 1.57 µg/cm^2^ (~0.5 µCi) of NAP (n = 4), 7.76 µg/cm^2^ (~1.5 µCi) of PHEN (n = 5), 35.96 µg/cm^2^ (~2.5 µCi) of BAP (n = 5), and 1.96 µg/cm^2^ (~1.0 µCi) of OPP (n = 5). The doses were aimed to apply finite doses to the skin surface based on OECD Guideline 428 for Testing of Chemicals in Skin Absorption: In Vitro finite doses (<5 mg/cm^2^) [45]. After dosing, diffusion cells were covered with Parafilm^®^ pieces (Pechiney Plastic Packaging, IL, USA) to minimize the loss of semi-volatile compounds.

### 2.4. Recovery of PAHs in Stratum Corneum and Skin Layers

The recovery of all compounds was good: 95.9 ± 1.2, 97.9 ± 3.5, and 99.0 ± 1.1% dose for naphthalene, phenanthrene, and benzo[a]pyrene. The majority of the dose that was not absorbed was recovered from the surface of the skin. Meanwhile, small amounts of naphthalene and benzo[a]pyrene were recovered from the skin itself, 0.7 ± 0.2 and 0.5 ± 0.5% dose, respectively. Phenanthrene had significantly more dose recovered from the skin, 32.3 ± 5.2% dose, than naphthalene and benzo[a]pyrene. Minimal amounts of each compound were recovered from the stratum corneum of the skin: 0.1 ± 0.1, 2.3 ± 0.6, and 0.1 ± 0.0 for naphthalene, phenanthrene, and benzo[a]pyrene, respectively. A summary of each chemical recovered from the skin can be found in Table 2.

### 2.5. Sample Analysis

Perfusate samples were taken at time points of 0, 15, 30, 45, 60, 75, 90, 120, 180, 240, 360, and 480 min after dosing. After the experiment, aliquots of the perfusate were transferred to new scintillation vials along with 15 mL of BioScint (National Diagnostics, GA, USA) and analyzed using a liquid scintillation counter for ^14^C determination. At the end of the experiment, the remaining dose was removed from the surface of the skin membrane with a cotton swab. The skin membranes were transferred to wax paper, where the surface of each skin disk was then tape-stripped (Scotch Tape; 3M, St. Paul, MN, USA) six times, placing three strips into a single scintillation vial, and adding 10 mL of ethyl acetate. After tape-stripping, the center of the skin disks was punched with an 8 mm biopsy tool, and the center and peripheral skins were separated and placed into individual scintillation vials along with 2 mL of BioSol (National Diagnostics, GA, USA). The skin samples were incubated at 50 °C for 8–12 h and analyzed using a liquid scintillation counter for ^14^C determination (MDL: 51 CPM). The fingertips of the gloves used during swabbing and tape stripping were extracted with ethanol.

### 2.6. Absorption Parameters Calculations

Absorption was defined as the total percentage of dose detected in the perfusate. Cumulative absorption (µg/cm^2^) was calculated by summing the total dose that was detected in the perfusate at each sampling time. Flux (µg/cm^2^/h) was obtained from the steady-state slope of the curves of cumulative absorption versus time, illustrated in Figure 1. The permeability coefficient (K_p_) (cm/h) was calculated from the ratio of the flux (µg/cm^2^/h) to the concentration (C_s_) (µg/cm^3^) of the dose. The dose concentration was obtained from 10 µL pre- and post-dose checks. The lag time (τ) was obtained by extrapolating the steady-state portion of the curve back to the time- or *x*-axis. This lag time was related back to diffusivity (D) and membrane thickness (L) by the following equation: D = L^2^/6 (τ). Student’s *t*-test was performed to determine significant differences at *p* < 0.05.

## 3. Results

### Dermal Absorption, Flux, Permeability, and Diffusivity

Both measures of total mass absorbed, and percent dose absorbed are valuable to understand different aspects of absorption kinetics and mechanisms for absorption [46]. Total mass absorbed indicates the amount that reaches systemic circulation, whereas percent dose absorbed provides information about the efficiency of the absorption process. The relationship between total mass absorbed and percent dose absorbed can provide insights into absorption at low or high doses [47,48]. Dermal absorption characteristics cumulative absorption (% dose), cumulative absorption (µg/cm^2^), dose absorption efficiency (% dose), lag time (minutes), diffusivity (cm^2^/h), and permeability (cm/h) are reported as mean ± standard deviation, while max flux (% dose/h) is reported as a range of the maximum value across measurements.

The cumulative absorption of ortho-phenylphenol was 30.1 ± 9.5% dose and is comparable to the absorption reported from a human volunteer study conducted by Timchalk et al. (1998), where the % dose of OPP recovered in urine was 42.7 ± 11.0 [49]. This confirms the accuracy of the flow-through system, as OPP behaved as expected.

The absorption of the three PAHs tested demonstrates that lower-molecular-weight PAHs penetrate more than larger-molecular-weight PAHs, which is consistent with previous findings. The max flux of naphthalene (7–11% dose/h) was greatest, followed by phenanthrene (1–2% dose/h), and benzo[a]pyrene had the lowest max flux among the PAH compounds (0.002–0.020% dose/h), and OPP was between NAP and PHEN (4–7% dose/h), as shown in Figure 2. The same trend was observed in dose absorption efficiency. Naphthalene (35.0 ± 4.6% dose) had the greatest amount of dose absorbed, followed by phenanthrene (6.8 ± 3.2% dose), and benzo[a]pyrene (0.03 ± 0.04% dose) had minimal amounts of absorption. The cumulative absorption (µg/cm^2^) of naphthalene, phenanthrene, and benzo[a]pyrene was 0.55 ± 0.07, 0.53 ± 0.25, and 0.011 ± 0.013, respectively. The cumulative absorption profiles of the PAH compounds can be seen in Figure 3. There were no significant differences in the mass absorbed between naphthalene and phenanthrene even though there was a significant difference in the percentage dose absorbed. This difference is due to the molecular weight of the compounds. Phenanthrene is larger, so although less of it was absorbed, it had a similar mass absorbed relative to naphthalene. While lower-molecular-weight PAHs are more likely to penetrate the skin, they are often not as toxic as the larger-molecular-weight PAHs. However, in this circumstance, NAP is a group 2B carcinogen (possibly carcinogenic to humans), while PHEN is a group 3 carcinogen (not classifiable as to its carcinogenicity to humans) [50]. NAP is more likely to penetrate the skin due to its size and cause dermal toxic effects, such as erythema and fissuring, and when absorbed, it leads to toxic system effects [51].

The lag time of naphthalene and phenanthrene was 52 ± 8 and 183 ± 21 min, respectively. The diffusivity (cm^2^/h) of naphthalene was greater than phenanthrene, 0.93 ± 0.15 vs. 0.22 ± 0.03. The same trend was observed in the permeability value (cm/h): 12.4 ± 2.2 × 10^−3^ for naphthalene and 2.1 ± 0.9 × 10^−3^ for phenanthrene.

Due to the low absorption of benzo[a]pyrene (<0.5% dose), it was difficult to accurately and confidently calculate the steady state. Without an accurate calculation of the steady state, the lag time, diffusivity, and permeability values of benzo[a]pyrene would also be inaccurate. Therefore, benzo[a]pyrene was excluded from the graphical illustration, absorption value calculations, and comparisons.

## 4. Discussion

### 4.1. Dermal Absorption Factors

Dermal absorption is a relatively simple concept of diffusion but is complicated by numerous chemical, skin, and environmental factors, and subsequent interactions [15]. In occupational exposure assessments, the job complexity and interindividual differences add another layer of complexity [1]. In general terms, for a chemical to penetrate the skin, it must first come in contact with the skin, often delivered by a vehicle. For firefighters, the vehicle for chemical exposure is dust, smoke, and particulate matter. Sometimes, no vehicle is used, as in the case of gases and vapors. The first step of dermal absorption is the partition of the chemical from the vehicle to the stratum corneum [52]. More lipophilic compounds will more easily partition into the stratum corneum than hydrophilic compounds because the stratum corneum is primarily a lipophilic environment comprising corneocytes and a lipid matrix [53]. After partitioning the stratum corneum, the second stage of dermal absorption begins, where the compound may penetrate the subsequent layers of the epidermis and dermis [54]. If the chemical penetrates these layers, it will likely be absorbed into the body by the capillary loops at the epidermis–dermis junction [55]. However, that a chemical penetrates the stratum corneum (SC) does not guarantee its absorption into the body; as Bourgart et al. (2019) and Liu et al. (2011) demonstrated, highly lipophilic chemicals are favorable to remain in the subcutaneous membrane without further diffusion [56,57]. Retained toxic chemicals within the skin may become systemically available later and can lead to unfavorable health effects in the site of absorption and organs. Highly lipophilic chemicals may more easily partition into the SC but struggle to navigate into viable skin, which is highly hydrophilic, confirmed by Roy et al. (1998), who reported a decrease in viable skin diffusion for compounds with log P greater than 2 [53].

Several factors can impact the absorption of a chemical, including dose vehicle; physicochemical properties such as size, molecular weight, and lipophilicity; environmental effects such as temperature and relative humidity; as well as skin conditions such as age, hydration, and anatomical location [58,59]. All of these variables can influence the bioavailability of the penetrating chemical and subsequent absorption. The Dalton 500 rule is a general predictor of a chemical’s ability to penetrate the skin. It states that chemicals with a molecular weight less than 500 Dalton can easily pass through the stratum corneum, but as the molecular weight of a chemical increases, their absorption rapidly declines [60]. All the PAH compounds tested have molecular weights less than 500 Dalton. Their absorption did follow the 500 Dalton trend, where the smallest PAH, naphthalene, had the greatest degree of absorption, followed by the middle-sized PAH phenanthrene, and then the largest PAH benzo[a]pyrene, which had the lowest amount of dose absorbed. In addition to size and molecular weight, the octanol–water partition coefficient or lipophilicity is a popular indicator of a chemical’s ability to penetrate the stratum corneum [61]. Chemicals with a log K_ow_ greater than -1 and less than 4 are expected to have high rates of absorption, and chemicals outside the previously mentioned range are expected to have approximately 10% absorption [46]. As the log K_ow_ of the PAHs in this experiment increased, the percentage dose absorbed decreased. The log K_ow_ value of naphthalene is 3.30, phenanthrene is 4.46, and benzo[a]pyrene is 6.13. Phenanthrene had a significantly lower percentage dose absorbed compared to naphthalene, and benzo[a]pyrene was even less. The drastic drop in absorption as lipophilicity increased could also be explained by the chemical’s solubility in the dose vehicle. The artificial sweat vehicle is primarily water with the addition of salts and amino acids. The solubility of the PAHs decreases rapidly in water, from 31 mg/L for naphthalene to 1.10 mg/L for phenanthrene to 0.00162 mg/L for benzo[a]pyrene. The low solubility of benzo[a]pyrene was recognized in the experimental design, so acetone was added to the dosing solution (1% of volume). Initial solubility tests indicated that the 1% acetone–sweat solution was viable for the experiment; however, this may have not been enough to prevent benzo[a]pyrene from “falling out of solution”. This was confirmed by the large amount of dose (>95% dose) recovered from the skin surface swabs post experiment, as shown in Table 2. The lack of dermal absorption (<1% dose absorbed) suggests that benzo[a]pyrene had limited to no contact with the skin, preventing dermal absorption.

The role of the dosing vehicle is to ensure that the drug or chemical of interest is uniformly distributed over the skin surface [46,62]. Organic solvents are typically the solvent of choice for dosing vehicles in dermal absorption studies because they quickly evaporate from the surface of the skin and leave the penetrating chemical on the surface of the skin, thus reducing any dose vehicle–chemical interactions [46]. Most dermal absorption studies on naphthalene and benzo[a]pyrene have used acetone or an acetone mixture for their dosing vehicles. However, the artificial sweat dosing vehicle used in this study was 99% water, and the 1% acetone added to the dose solution was not expected to impact diffusion. While the artificial sweat vehicle appeared to greatly impact the absorption of benzo[a]pyrene, the purpose of applying the PAH compounds in the artificial sweat dosing vehicle was to understand the dermal absorption during a worst-case scenario of a fireground contaminant coming into contact with a firefighter’s skin. The other two PAHs, NAP and PHEN, penetrated the skin in the presence of the aqueous environment, indicating little to no impact of sweat on dermal absorption. In this study, the trend of less lipophilic compounds having greater absorption is a result of the dose vehicle–chemical interaction. Furthermore, during the experiment, the diffusion cells were occluded, preventing the artificial sweat vehicle from evaporating, leading to increased interactions between the dosing vehicle and penetrating chemical.

### 4.2. Comparisons with Other Dermal Absorption Studies

#### 4.2.1. Naphthalene Absorption

The percutaneous absorption of naphthalene has been extensively studied in numerous dose vehicles, including powder, saturated aqueous solutions, jet fuel mixtures, and organic solvent mixtures. Baynes et al. (2000) studied the percutaneous absorption of NAP from various jet fuels and reported flux values of 0.156 ± 0.03 µg/cm^2^/h [63]. Frasch et al. (2007) studied the absorption of NAP in a powder and saturated aqueous vehicle. The steady state flux of NAP decreased by 45% from the powder (30.39 ± 2.03 µg/cm^2^/h) to the aqueous vehicle (13.61 ± 4.54 µg/cm^2^/h) [64]. The flux of NAP in the aqueous vehicle from Frederick’s study was significantly higher than the flux values in this study and that of Baynes et al. These differences are likely a result of the infinite dose applied and the different dose vehicles. The flux value of NAP in this study was 0.124 ± 0.02 µg/cm^2^/h similar to the flux value in Baynes et al. (2000) and significantly less than the value reported in Frasch et al. (2007). However, Frasch et al. (2007) used an infinite dose, whereas this study used a finite dose, which is the likely cause for the differences. Although the dose conditions were different, the lag times were similar: 1.0 ± 0.4 h vs. 0.9 ± 0.2 h in Frederick et al.’s (2007) study and this study, respectively. In all studies, NAP was demonstrated to readily penetrate the skin in lipophilic and hydrophilic environments.

#### 4.2.2. Phenanthrene Absorption

The percutaneous absorption of phenanthrene has not been studied to the same extent as naphthalene and beno[a]pyrene, as its properties fall between formerly mentioned PAH compounds. Ng et al. (1990) was one of the earliest studies on percutaneous absorption of PHEN, dosing 6.6 µg/cm^2^ in an acetone vehicle, demonstrating increased absorption at the 6 h mark from 2.4% to 38.9% when changing the receptor fluid from HHBSS to HHBSS + 4% bovine serum albumin (BSA) in guinea pig skin [65]. Sartorelli et al. (2001) showed 9.58 ± 4.70% dose absorption of phenanthrene at 6 h with an acetone vehicle in human cadaver skin [66]. Neither study reported flux, lag time, nor permeability coefficient. In this study, 6.8 ± 3.2% dose of the 7.8 µg/cm^2^ of PHEN applied to the skin was absorbed. Although on the lower end, the absorption in this study is within the range of absorption reported by Ng et al. (1990) and is consistent with the absorption reported by Sartorelli et al. (2001). The cumulative percentage dose of PHEN absorbed was significantly less than NAP, suggesting that PHEN is less efficient at navigating the skin barrier.

#### 4.2.3. Benzo[a]pyrene Absorption

The absorption results of BAP in this study were unlike results from previous in vivo and in vitro studies which have previously shown that benzo[a]pyrene readily penetrates the skin. Yang et al. (1986) reported BAP absorption in Sprague Dawley rats after 24 h in the range of 0.14 ± 0.04–17.2 ± 2.0% dose in vitro and 5.8% dose in vivo [38]. Moody et al. (1995) reported BAP % dose in the reception fluid after 48 h to be 27.5 ± 1.33, 3.3 ± 0.68, and 1.5 ± 0.63 in hairless guinea pig and 32-year-old and 50-year-old humans, respectively [26]. Kao et al. (1985) reported that the absorption of BAP after 17 h was less than 0.8 and 0.2 (% dose) in viable and nonviable rat skins, respectively [67]. Each previous study used an acetone or acetone mixture for their dose vehicle, whereas this study used an aqueous artificial sweat dose vehicle. Although the artificial sweat dosing vehicle is not a common vehicle for traditional dermal absorption studies, it does provide a more realistic representation of absorption for firefighters. The low absorption of BAP in this study is likely a dose vehicle–chemical interaction, as the solubility of BAP in water is low: 1.62 × 10^−3^ mg/L. This study shows that BAP absorption can be drastically reduced when the skin is saturated with sweat.

### 4.3. Application to Firefighters

An important consideration for dermal exposure for firefighters is that the primary vehicle for PAH exposure is particulate matter and vapor. Few studies have looked at the absorption of PAHs through a vapor or gaseous vehicle; meanwhile, there are some studies that have tested the absorption of PAHs in dust and soil, which are similar to the particulate matter that firefighters would encounter [24,39,66]. Wester et al. (1990) reported that the absorption of BAP from a soil vehicle is significantly lower than that from an acetone vehicle in human cadaver skin: 1.4 ± 0.9% vs. 23.7 ± 9.7%, respectively [39]. Sartorelli et al. (2001) later supported these results by showing the absorption of PAHs from an acetone vehicle and minimal absorption from coal dust [66]. Luo et al. (2020) later built upon Sartorelli’s study looking at the PAH absorption from indoor dust in a synthetic sweat and sebum mixture [24]. Luo et al.’s (2020) findings demonstrated that low-molecular-weight PAHs (NAP) had greater absorption than high-molecular-weight PAHs (BAP), which agree with the findings in this study. The purpose of using an artificial sweat solution as the dosing vehicle in this study was to investigate the impact of sweat on the absorption of PAHs.

Overall, firefighter dermal exposures are difficult to quantify. Although chemicals may be found on the skin, they may not be entirely available for dermal absorption. PAHs on the fireground are present in vapors, particulate matter, or smoke sources. Vehicles such as particulate matter or smoke that are primarily of carbon will retain any chemicals adsorbed to the surface of the particle due to van der Waals forces and analogous phenomena, thus decreasing their availability for absorption. Sartorelli et al. (2001) highlights the monolayer phenomenon, which states that only a monolayer of soil or particulates will be in contact with the surface of the skin and likely limit dermal uptake from solids [66]. However, sweat may play an important role in firefighters’ exposure. Luo et al.’s (2020) findings suggest that dermal bioaccessiblity of PAHs in indoor dust may change depending on the solid matter/liquid ratio, solubility, and sorption mechanism between organic matter in indoor dust and PAHs [24].

Another important consideration for firefighter dermal exposures is the time from exposure to decontamination. Many fire stations are now implementing on-scene decontamination strategies to reduce the time to a first pass of cleaning for the turnout gear and firefighters themselves [1]. A survey study of 482 firefighters highlighted that although firefighters had positive attitudes towards clean turnout gear and believed that cleaning their gear would reduce their risk of cancer, there was a large disconnect between their beliefs and actions. The survey results showed that showering within the hour of exposure was the only method of decontamination implemented routinely, and few firefighters, less than 20%, reported that they frequently or always cleaned their gear before leaving the fire scene or used decontamination wipes [68].

Skin decontamination has not been studied as well as turnout decontamination in the fire service industry. One study conducted by Keir et al. (2023) showed that water and water–soap decontamination strategies are capable of removing significant amounts of PAHs from the skin, although no significant reduction in the internal dose was observed [69]. For skin decontamination, it is commonly practiced to use copious amounts of water or water + soap to remove contaminants from the skin surface, and it is generally recommended for most contaminants [70]. Water and water + soap skin decontamination has been shown to decrease contamination levels and dermal absorption of pesticides, cleaning, cosmetic, and therapeutic agents, as well as chemical warfare agents and simulants [71,72,73]. However, there is concern among firefighters regarding the ingredients in skin cleansing wipes enhancing dermal absorption by the “wash-in” phenomenon. The “wash-in” phenomenon is when solvents enhance the penetration of chemicals rather than washing them off [26]. The “wash-in” phenomenon has been demonstrated in multiple studies where a soap wash was used to clean the skin [26,74,75].

The efficiency for the skin cleansing decontamination method can be partly explained by the physicochemical properties of each contaminant, such as hydrophobicity, octanol–water partition coefficient, molecular weight, and volatility [76]. Chemicals that readily penetrate the skin, like naphthalene, become unavailable to be washed off or removed after penetrating the skin. So, chemical properties that favor skin penetration are unfavorable for decontamination efficacy. Highly volatile compounds generally have reduced rates of dermal penetration due to their tendency to evaporate from the skin [77].

The lag times of naphthalene and phenanthrene are approximately 60 min and 180 min, respectively. This indicates a window of opportunity for firefighters to perform skin decontamination to reduce their exposures to PAHs. However, significant amounts (>30% dose) of PHEN were found on the skin deposits at the end of the eight-hour exposure. Compounds remaining in the skin compartments post exposure may continue to be absorbed through the skin [78]. Strong chemicals binding to skin components or the removal of the chemicals through skin shedding would prevent the systemic absorption of chemicals remaining in the SC and skin [73]. However, the remaining NAP and PHEN in the skin or skin compartments will continue to be absorbed post-exposure [79,80]. It is unlikely to remove contaminants that have already penetrated the skin; however, specialized skin decontamination methods such as dermal decontamination gels (DDgels) have been shown to back extract chemical warfare agents from the skin [70]. Unfortunately, firefighters may not have access to specialized decontamination methods, and water and soapy water skin decontamination is unlikely to remove any contaminants that remain on the skin.

Currently, firefighters are most likely to shower upon returning to the fire station, one to two hours post-exposure [68]. To minimize dermal exposures, firefighters should partake in water or soapy water skin decontamination as early as possible to remove the bulk of the contaminant from the skin. Multiple skin decontamination studies have emphasized that skin decontamination should occur as soon as possible (<10 min post-exposure) to remove the bulk of the contaminant from the skin surface and minimize post-exposure absorption [71,72,81]. The impact of skin decontamination would be most effective for compounds like PHEN and BAP, which are more likely to be on the particulate phase and be deposited on the skin, and also have lower rates of skin penetration in aqueous environments [5,82,83]. Immediate skin decontamination will be most effective for reducing these compounds, such as PHEN, on the skin and the amount remaining on the skin compartment post-exposure. However, the amount of reduction is difficult to predict as factors such as cleaning thoroughness, anatomical site, and skin decontamination method may change effectiveness. The time differences in current practice vs. what is recommended warrant investigation into the impact of time till skin decontamination for firefighters. Additionally, investigation into the “wash-in” phenomenon and the cost vs. benefit of skin decontamination for PAHs and other fireground contaminants is justified.

The goal of this study was to simulate the worst-case scenario of firefighter exposures, where firefighters are continuously exposed during an eight-hour emergency response or work shift. The artificial sweat dose vehicle was selected over traditional organic solvent vehicles due to the sweat firefighters produce during fire response. Athletes can lose as much as 0.5–2.0 L of sweat per hour, and while wearing turnout gear, firefighters are capable of equal, if not greater, amounts of sweat loss [84]. Although crucial information regarding the absorption of PAHs in artificial sweat was generated, there are limitations in this study. Liquid chemicals were used rather than vapor, particulate, or soil, which were chosen to due to their greater accessibility and the lack of radiolabeled vapor, particulate, or soil materials. Additionally, as previously mentioned, vehicles with higher carbon amounts resulted in decreased amounts of PAH absorption compared to organic-solvent vehicles [39]. Furthermore, in vitro studies have repeatedly reported lower absorption compared to in vivo studies across multiple species, including humans and pigs [26,38,39]. Although no animal skin model can replicate the absorption in human skin, pig or porcine skin is the most relevant animal model and has been reviewed and validated over many years [54,85]. Considering the limitations of this study, the results are valuable to better understanding firefighters’ chemical exposures during fire response and the potential to decrease chemical exposures with skin decontamination. Additionally, this is the first study to investigate the absorption of NAP, PHEN, and BAP in an artificial sweat dosing vehicle in vitro. This study is also the first to report lag time for PHEN in porcine skin and to demonstrate that the absorption of BAP is drastically reduced in an aqueous environment such as artificial sweat.

## 5. Conclusions

Exposure to fireground contaminants is inevitable for firefighters, whether it be exposure during fire response activities, overhaul, or post-fire exposures. This study aimed to evaluate the dermal absorption of PAHs in vitro for firefighters by using an artificial sweat dosing vehicle to mimic the sweat produced during fire response. Molecular weight and solubility appeared to play the largest role in PAH absorption when the skin was saturated with sweat. Naphthalene had the highest percent dose absorbed and was the smallest PAH and the most soluble in water. Phenanthrene is a medium-sized PAH and is less soluble in water than naphthalene; it had a lower percentage dose absorbed. Lastly, benzo[a]pyrene, the largest and least soluble PAH, had minimal amounts absorbed in the skin: less than 1% total dose.

On-scene decontamination is becoming more popular among firefighters. One form of on-scene decontamination is skin decontamination. The lag time of naphthalene and phenanthrene and the minimal absorption of benzo[a]pyrene in the presence of sweat indicates that there is a window of opportunity for firefighters to remove contaminants from their skin and reduce the post-exposure absorption of contaminants. The goal of skin decontamination for firefighters should be to remove the bulk of contaminants on the skin to minimize post-exposure absorption and overall dermal exposures.

Lastly, chemical absorption from carbon-based vehicles, such as particulate matter from smoke and soot, is lower compared to either aqueous or organic solvents. This study used liquid chemicals, and it is suggested that if the study is repeated with a particulate or soil vehicle, then the absorption of PAHs would be reduced. The role of the dose vehicle is critical for firefighters to consider as their exposures include a wide range of toxic chemicals in various phases of matter.

## Figures and Tables

**Figure 1 toxics-12-00588-f001:**
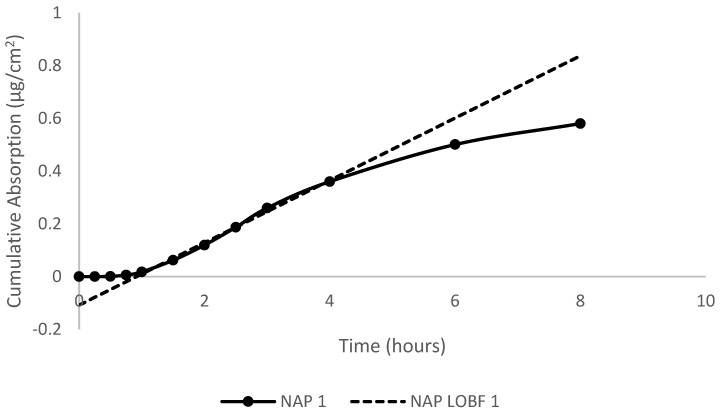
Plot of cumulative absorption (µg/cm^2^) versus time (h) for naphthalene in artificial sweat following topical application to porcine skin in in vitro flow-through diffusion cells. The best-fit line was used to calculate the steady-state flux for the test compounds.

**Figure 2 toxics-12-00588-f002:**
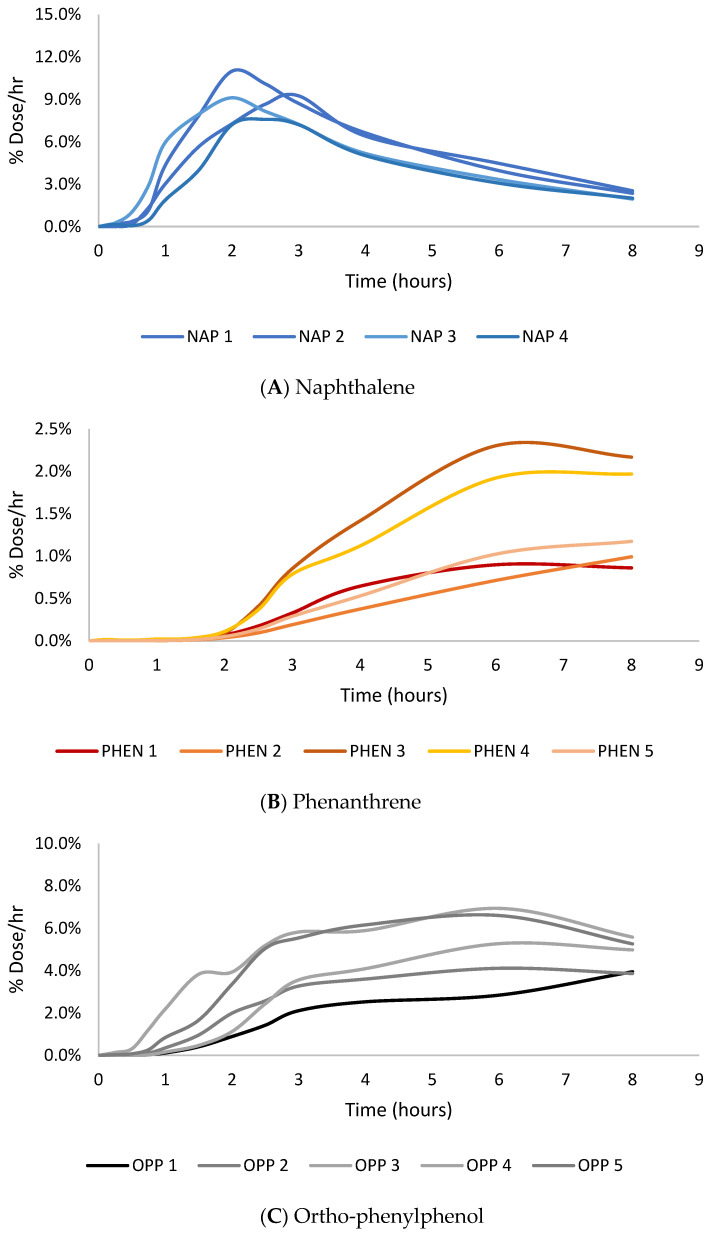
Peak flux (% dose/h) vs. time (hours) absorption profiles for naphthalene (**A**), phenanthrene (**B**), and ortho-phenylphenol (**C**).

**Figure 3 toxics-12-00588-f003:**
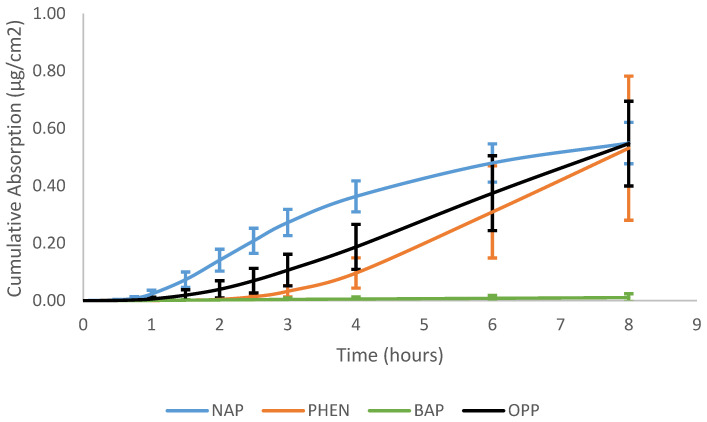
Cumulative absorption (µg/cm^2^) vs. time (hours) profiles of naphthalene, phenanthrene, benzo[a]pyrene, and ortho-phenylphenol in porcine skin.

**Table 1 toxics-12-00588-t001:** Chemical properties of radiolabeled compounds used in flow-through experiments.

	Naphthalene	Phenanthrene	Benzo[a]pyrene	Orthophenylphenol
Formula	C_10_H_8_	C_14_H_10_	C_20_H_12_	C_12_H_10_O
CAS Number	91-20-3	85-01-8	50-32-8	90-43-7
IARC Classification	2B	3	1	2B
Molecular Weight	128.2	178.2	252.3	170.2
Number of Aromatic Rings	2	3	5	2
Log K_ow_	3.30 ^a^	4.46 ^a^	6.13 ^a^	3.09 ^a^
Solubility in Water at 25 °C (mg/L)	31 ^a^	1.10 ^a^	1.62 × 10^−3 a^	700 ^a^
Radioactivity	1-^14^C	9-^14^C	7-^14^C	Ring-^14^C
Specific Activity (mCi/mmol)	57	55	27	150
Concentration (mCi/mL)	0.1	0.1	0.1	0.1
Solvent	Ethanol	Ethanol	Toluene	Ethanol

Values obtained from ^a^ Hazardous Substances Data Bank IARC Classifications: 1-Known Carcinogen, 2A-Probably Carcinogenic to humans, 2B-Possibly Carcinogenic to humans, 3-Not classifiable as to its carcinogenicity to humans, 4-Probably not carcinogenic to humans.

**Table 2 toxics-12-00588-t002:** Mass balance summary of ^14^C PAH compounds.

	Dose (µg/cm^2^)	Skin Surface (% Dose)	Stratum Corneum (% Dose)	Skin (% Dose)	Absorption (% Dose)	Total Recovery (% Dose)
Naphthalene	1.57	60.3 ± 4.5	0.1 ± 0.1	0.7 ± 0.2	35.0 ± 4.6	96.1 ± 1.1
Phenanthrene	7.76	56.5 ± 3.5	2.3 ± 0.6	32.4 ± 5.2	6.8 ± 3.2	98.1 ± 3.3
Benzo[a]pyrene	35.96	98.0 ± 2.1	0.1 ± 0.2	1.8 ± 1.8	0.03 ± 0.04	100.0 ± 2.2
Ortho-phenylphenol	1.96	56.4 ± 11.7	0.3 ± 0.3	10.6 ± 3.3	30.1 ± 9.5	97.4 ± 3.0

## Data Availability

The data used to generate tables, figures, and absorption values presented in this study can be found in Appendix A.

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
