# Peer review of "Percutaneous Absorption of Fireground Contaminants: Naphthalene, Phenanthrene, and Benzo[a]pyrene in Porcine Skin in an Artificial Sweat Vehicle"

_toxics, 2024, doi:10.3390/toxics12080588_

Round 1

Reviewer 1 Report

Comments and Suggestions for Authors

1. You suggest immediate skin cleansing practices implemented. But how efficient is skin cleansing? What does literature tell, and will efficiency depend on the chemicals (size, lipophilicity) etc.? please inform us.

2.   Lag-time tells when the chemical has penetrated through the skin and can be measured in the receptor fluid. At this point, however, a significant dose may reside within the skin compartment. Please reflect on the potential for this temporary skin deposition to be removed by skin cleansing or to slowly continue being absorbed. Should chemicals deposited in the skin department be counted as available for subsequent further permeation?

3. Please reflect and check if literature is available on the distribution between larger and smaller PAHs in true exposure scenarios, i.e. exposure assessments of fire fighters. This could also be carried over to the discussion on the actual risk reducing potential suggested through immediate skin cleansing.

4.        Line 123: which anatomical site of the pigs was used for this study? Please include information.

5.        Line 128: integrity testing – well, well, well – but testing integrity is also testing whether skin was damaged during mounting procedures. Replacing compromised cells reduces experimental variability.

6.        Line 132: what is the argument for using this flow rate?

7.        Dosing: did you aim for finite or infinite dosing conditions, which could be relevant for subsequent calculation of Kp based on penetration curves?

8.        Line 190-191 (no differences in the mass absorbed..): to what extent is this a relevant observation? Is it the weight of the absorbed chemicals or the number of molecules that is most relevant for the toxicity, i.e. binding to DNA?

9.        Line 215-216: very nice to see that the authors did this reassuring calculation and included it.

10.   Table 2: what is the potential for removing the large skin deposition (32.4% of the dose) through skin cleansing procedures? The importance of this question is also dependent on the size distribution of PAHs in the real-life exposure. Please reflect.

11.   Line 240-248: please relate this discussion more closely to your own findings.

12.   Line 274-275: how could you easily have checked this? Where would you expect BaP to end up if it was ‘falling out of solution’?

13.   Line 338-361:please reflect on how your own experimental/methodological approach fit in in this discussion?

14.   I miss a semi-short paragraph in the discussion reflecting on strengths and weaknesses in your study.

Author Response

Please see the attachment for responses to Reviewer 1 comments

Reviewer 2 Report

Comments and Suggestions for Authors

Title: Percutaneous Absorption of Fireground Contaminants: Naphthalene, Phenanthrene, and Benzo[a]pyrene in Porcine Skin in an Artificial Sweat Vehicle

In the study dermal absorption of naphthalene, phenanthrene, and benzo[a]pyrene using porcine skin for understanding the firefighters’ dermal exposures were studied. Multiple absorption characteristics (cumulative absorption, percent dose absorbed, diffusivity, flux, lag time, and permeability) were examined, which was interesting and showed the impact of the study more clearly. The research purpose is clearly stated and the manuscript is well-written. In my opinion, the manuscript must be focused on that line of descriptions of analytical methods and could be considered after major revisions. 

  1. The references are inadequate. For example, Lines 37-38, Lines 41-44, Lines 46-53, Liness 201-204, Lines 228-241, Lines 250-255, Lines 257-263, Lines 279-280, Lines 283-292, Lines 299-306, Lines 308-312, Lines 338-350, Lines 352-359 and Lines 368-374.
  2. The quality of the figures should be improved. The scale is required.
  3. Lines 166 what does that mean? “ ErrorReference source not found”
  4. Multiple Figures 1 are presented.
  5. The details of dermal absorption of naphthalene, phenanthrene, and benzo[a]pyrene should be presented in the Method section. The authors should state the initial exposure concentration of naphthalene, phenanthrene, and benzo[a]pyrene in dermal absorption. Furthermore, the detection limits, linearity, and repeatability of the measurements in naphthalene, phenanthrene, and benzo[a]pyrene using a 14C liquid scintillation counter should be included in the manuscript.
  6. Table 2. The descriptions of the experiment details of each chemical recovered from the skin should be illustrated in the Method section.

Author Response

Please see the attachment for response to Reviewer 2 comments

Round 2

Reviewer 1 Report

Comments and Suggestions for Authors

I find that the authors have addressed my questions in a fair way.

Rewriting large parts of the discussion section allowed for relevant perspectives to be included (though the discussion has become rather long).

No further requests from my side.

The interesting, yet not toxicological, challenge is how to get the firefighters to clean their gear and themself within the initial hour following action.

Reviewer 2 Report

Comments and Suggestions for Authors

Accept